# Real-World Urban Light Emission Functions and Quantitative Comparison with Spacecraft Measurements

Brian R. Espey * , Xinhang Yan and Kevin Patrascu

School of Physics, Trinity College Dublin, University of Dublin, College Green, Dublin 2, Ireland
* Correspondence: brian.espey@tcd.ie

**Abstract:** We provide quantitative results from GIS-based modelling of urban emission functions for a range of representative low- and mid-rise locations, ranging from individual streets to residential communities within cities, as well as entire towns and city regions. Our general aim is to determine whether lantern photometry or built environment has the dominant effect on light pollution and whether it is possible to derive a common emission function applicable to regions of similar type. We demonstrate the scalability of our work by providing results for the largest urban area modelled to date, comprising the central 117 km$^2$ area of Dublin City and containing nearly 42,000 public lights. Our results show a general similarity in the shape of the azimuthally averaged emission function for all areas examined, with differences in the angular distribution of total light output depending primarily on the nature of the lighting and, to a smaller extent, on the obscuring environment, including seasonal foliage effects. Our results are also consistent with the emission function derived from the inversion of worldwide skyglow data, supporting our general results by an independent method. Additionally, a comparison with global satellite observations shows that our results are consistent with the deduced angular emission function for other low-rise areas worldwide. Finally, we validate our approach by demonstrating very good agreement between our results and calibrated imagery taken from the International Space Station of a range of residential locations. To our knowledge, this is the first such detailed quantitative verification of light loss calculations and supports the underlying assumptions of the emission function model. Based on our findings, we conclude that it should be possible to apply our approach more generally to produce estimates of the energy and environmental impact of urban areas, which can be applied in a statistical sense. However, more accurate values will depend on the details of the particular locations and require treatment of atmospheric scattering, as well as differences in the spectral nature of the sources.

**Keywords:** light pollution; public lighting; photometry; GIS; LiDAR; digital elevation models; SUOMI NPP; VIIRS; DNB

## 1. Introduction

Our general goal in this work is to address a number of topics of relevance to energy and light pollution measurement. The emission function, i.e., the amount of light emitted in different directions, is, a priori, unknown for complex situations involving different numbers and types of lighting in urban environments comprised of opaque and reflective surfaces. As a result, it needs to be determined via analysis of ground-based scattering observations [1–3], or by means of theoretical models using either analytic simplifications or more detailed models requiring complex and time-consuming, high-performance computing approaches, e.g., [4–7]. As a particular example of a worldwide approach to modelling, Falchi et al. developed a globally representative emission function based on their study of zenithal skyglow measurements taken around and outside a sample of urban areas to derive a representative emission function which they then used to predict light pollution at sites remote from the emitting source [8].

As discussed in our first paper [9]—hereafter Paper One—our method to obtain the emission function involves an innovative, semi-empirical, GIS-based approach. In this approach, we model the emission data from the ground upwards using information about public lighting, including the angular photometry of the individual lanterns, as well as elevation data at 1–2 m spatial resolution to provide detailed information on obstructions, and we refer the reader to Paper One for background details. The output of this approach is a representative emission function which includes both direct and diffuse emission and can be readily tailored to account for differences in light sources, surface reflectivity, etc. Although GIS-based approaches using estimated light locations coupled with ground measurements of illumination have been attempted previously to our knowledge there has been no integration of both detailed lighting and obstruction information to produce a comprehensive picture for entire areas—see, e.g., [10].

In this paper, we develop our work with applications to specific urban areas of increasing area and complexity to produce representative emission functions, i.e., a general description of how much light is emitted at differing azimuth and zenith angles. Such emission functions have utility as they can be used as input to radiative transfer models of atmospheric transmission and scattering to model light pollution's impact on the urban areas themselves, as well as on the wider environment. Additionally, this approach can be used to provide a means to interpret satellite observations. Our specific aim in this paper is to study a range of representative residential and general urban areas to determine their emission functions and assess the relative importance of lighting photometry compared with areal geometry, i.e., the relative importance of the number and distribution of public lights compared with the number and type of buildings or trees. We note that Irish building heights are relatively low and predominantly in the low- to mid-rise class, as there is a 26 m high limit in the centre of Dublin, with lower levels in suburban areas and outside the capital. In the areas of study chosen for this paper, building heights are typically two to three stories.

## 2. Materials and Methods

### 2.1. Selection of Areas

We have sampled a range of local environments classified as having low- to mid-rise buildings and also within the categories of the continuous or discontinuous urban fabric as defined by the EU Corine land use categorisation. To span a range of representative environments, we have chosen areas ranging from suburban residential, through entire towns, to city centre locations. We include the areas already introduced in Paper One together with a range of new locations and apply our study to additional elevation datasets, which have since become available. Note that our models only include public lighting as the information from local authorities enables us to have a complete and detailed inventory to metre-level accuracy, though we aim to extend our work in the future to include other light sources. The date range modelled is the period 2015–2017, as this is the epoch covered by our public lighting databases, as well as within a few years from when the digital elevation data were obtained. For this period, public lighting was predominantly of the low- or high-pressure sodium type, which we will refer to as LPS and HPS, respectively.

### 2.2. Data and Locations

Our work makes use of a number of input datasets, including light detection and ranging (LiDAR) digital elevation datasets, which map buildings and trees, and we couple this with information on public lighting types and locations, all at metre-scale precision. Basic information regarding the digital elevation datasets is given in Table 1, and information regarding the source of the datasets is provided in the Data Availability section. In subsequent sections, we will discuss the test regions in rough order of their size, from individual residential areas to towns and larger city areas.

**Table 1.** LiDAR DEM datasets used in this work. Data generally have a vertical rms accuracy of 20 cm. Note that datasets cover a range of seasons.

| Origin | Spatial Resolution | Area | Date Obtained |
|---|---|---|---|
| NYU | 1 m | Dublin Central | March 2015 |
| Commercial | 1 m | Dublin | May–June 2018 |
| OPW | 1 m | Cork Towns | November–December 2015 |
| OPW | 2 m | Newport, Westport, Tralee | August 2012 |
| OPW | 2 m | Dublin | December 2011–February 2012 |

In Table 2 below, we present a summary of the characteristics for all areas studied in this paper: "Residential" areas are suburban areas in Dublin and are detailed in Section 3.1 below. "Cork towns" refers to the town subset reported in Paper One, and further information is provided there. In all models, we use a surface reflectivity of 10%, representative of tarred (asphalt) urban roadways [11]. In the basic information tabulated for each location, included are a number of parameters related to light output, including that labelled "% S/W" which gives the ratio of summer to winter output and so provides an indication of the relative impact of foliage on the total light emitted to the upward hemisphere. The last three columns provide data for the 2 m resolution (winter) data. "Direct/Total" indicates the relative importance of direct emission to the upward hemisphere to the diffuse component due to light reflected from the ground and structures. The column labelled "% to zenith" indicates the light lost due to obstructions for the case of the winter model. Finally, the "Lamfit" column provides summaries of the total output as estimated by integrating a Lambertian fit to a series of near-nadir (i.e., observations within 20° of nadir) observations, i.e., from the point-of-view of air- or space-borne observations. The intention of this column is to indicate the potential error introduced by the common (daytime) remote sensing assumption that all emission is purely due to Lambertian (diffuse) emission. For comparison, the equivalent correction factor to a Lambertian approximation for the model adopted by Falchi et al. [8] is 12% larger than the Lambertian integral.

**Table 2.** Summary table of the parameters for a range of areas reported in this paper as derived from our model fits. For a description of the column headings, see the text.

| Location | Pixel Size | No. Areas | Area (km²) | No. Lights Modelled | % LPS (Median) | % S/W | Direct/Total (Median) | % to Zenith (Median) | Lamfit (Median) |
|---|---|---|---|---|---|---|---|---|---|
| Residential LPS 55 W | 1 m, 2 m | 5 | 0.04–0.2 | 12–112 | 99% | 76% | 71% | 80% | 2.9 |
| Residential other | 1 m, 2 m | 6 | 0.02–0.06 | 7–26 | 100% | 77% | 14% | 86% | 1.1 |
| Cork towns | 1 m | 9 | 0.3–3.9 | 9–149 | 35% | a | 21% | 77% | 1.2 |
| Newport Westport Tralee | 2 m | 3 | 1 8 19 | 135 1098 5121 | 0% 13% 100% | a | 9% | 73% | 1.1 |
| Dublin NYU | 1 m | 1 | 1.5 | 1012 | 13% | 100% | 24% | 59% | 1.3 |
| DCC | 1 m, 2 m | 1 | 117 | 41,763 | 51% | 89% | b | 80% | 1.0 |

(a) Only one DSM dataset was available for these areas, so no comparison could be made. (b) DCC area was modelled without direct emission component.

The last two columns can be combined to estimate the light output from near-nadir satellite observations. For instance, in the case of "Residential other" and "DCC" areas, a

Lambertian model would have to be increased by a factor of Lamfit/% to zenith, i.e., by approximately 30% to obtain the total light output. For the LPS 55 W-dominated residential areas, a similar model would have to be scaled by a factor of 3.6, which is a sizable correction due to the poorer light control of these older prismatic lens lanterns. For the city centre areas, on the other hand, lighting is dominated by HPS units along the main roadways, with old-style lanterns bordering park areas where trees are located which help block emissions. Outside of these park areas, there are relatively few large trees in the inner-city area and this, together with more powerful HPS lanterns installed around 8–10 m above road level, results in these lights with their better lighting control dominating the total emission.

## 3. Results

### 3.1. Dublin Residential Areas

For the residential areas, we chose locations where various types of LPS lighting dominated. This is because these older units have the widest distribution of light, and so the largest proportion of direct to diffuse emission, with up to 10% of the lamp output being directly emitted above the horizontal for LPS 55 W prismatic lens units [9]. Our aim was also to model locations which can be readily compared with spacecraft data, and the relatively monochromatic light provided by LPS units facilitates calculations to convert from lumens to watts, which are the more standard units for earth observation (see also Section 3.7).

We used colour imagery obtained from the International Space Station (ISS) in conjunction with our public lighting databases to identify contiguous areas which contained LPS lighting with similar wattage predominantly throughout, either 55 W, 90 W, or 135 W, depending on location. In a search of the NASA archive we located image ISS045-E-170140 from 2015, which is roughly contemporaneous with our lighting database and elevation data. We also used this image to identify areas which were uncontaminated by neighbouring lighting, such as commercial or architectural floodlighting. In this regard, the ability to determine both intensity and colour from the ISS camera data proved useful in distinguishing useful areas, as these locations are easily distinguishable in the imagery and—for the wattages used in our study—at intensities where the camera response is linear. Our chosen areas lie at distances ranging from two to nine kilometres from the centre of Dublin, with the bulk being beyond four kilometres.

The representative residential areas chosen were sufficiently large to contain tens to just over one hundred lanterns (see Table 2). In terms of the general environment, residential streets had roadways 6 to 8 m wide, with tree-lined pavements and an average building height (ABH)—defined as height above roadway of 3 m or more, to exclude walls, shrubs, etc.—of 7.1 m, with a standard deviation (SDBH) of 2.36 m. The residential public lighting lanterns were typically located at 6 to 7 m above the ground, while higher wattage lighting was located at 8 to 10 m above the roadway on busier roads which were up to 11 m wide. The geometry of the individual areas varied from housing estates to single streets and are classed as discontinuous urban fabric in the EU Corine 2018 land cover database. Modelling of the areas to produce azimuthally averaged emission functions followed the procedure outlined in Paper One, and we refer the reader to that paper for details. A point to note for all results reported in this work is that, although the lantern photometry only provides data for a discrete set of angles, we have fitted a sixth-order polynomial to the calculated values in order to produce a smooth curve for both display and also numerical integration purposes.

The results presented in Figure 1 show the azimuthally averaged radiant intensity emission functions. Each curve is normalised to its zenithal value so that the relative behaviour of the curves can be readily seen with the data falling into two broad categories, dependent on the nature of the lantern. The curves with the highest emission correspond to locations where there is a large component of emission to intermediate angles from 55 W LPS lanterns with prism lenses. The local environment is a secondary effect which

generates the spread of results within both broad categories, though the direct emission from the 55 W lanterns exaggerates this behaviour.

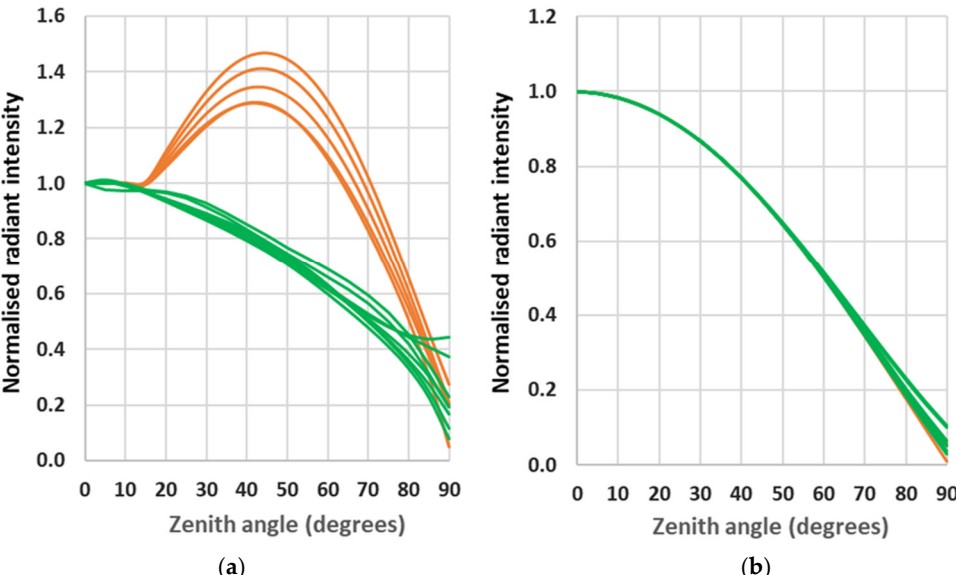

**Figure 1.** Azimuthally averaged radiant intensity results for the Dublin residential areas containing LPS lighting with plots in orange showing the results for the "Residential LPS 55 W" areas, and those in green showing the "Residential other" areas (see Table 2). In order to compare the areas, the radiances have been normalised to the zenithal emission in each case. Subfigure (**a**) shows the total (diffuse + direct) emission for areas containing primarily 55 W units with prismatic lenses, which contribute to excess emission in the upper hemisphere. Subfigure (**b**) shows the excellent agreement between all areas when only the diffuse (reflected) component is considered. For references to colour, see the on-line version of the text (Supplementary Materials).

To illustrate the importance of direct emission in these cases, we show in Figure 1b the reflected component only, which illustrates that there is little difference between any of the areas in terms of environmental modification of emission. As a side note, this plot also indicates the expected behaviour when older prismatic lanterns are replaced with more modern units with better light control, where less light is emitted in directions where it misses the ground. The contribution of this light can also be inferred from Table 2, where the integrated light ratio is shown under the column "Direct/Total".

*3.2. Irish Towns*

We also modelled entire rural Irish towns, comprising the towns in the Cork area introduced in Paper One supplemented by larger towns. These towns were selected on the basis of available digital elevation models, whose details are presented in Table 2. As in the case of the residential areas, Irish towns are generally low- to mid-rise in height with a mix of street widths: for market towns, there is a wider street or square, but narrower streets elsewhere. For these locations, the lighting is again predominantly of LPS and HPS type, though not as uniform in wattage as the Dublin residential areas, which were chosen to be near uniform in character. Additionally, when moving to these larger towns, there may be a range of road sizes and also a larger contribution from commercial lighting. The current work does not intend to provide a complete model for these towns in terms of all lighting components, but rather to test our model over larger areas with more numerous lights. The intended goal is to determine whether there are any commonalities in terms of emission function for this range of locations and to study how differences in the nature of the wider urban environment may influence the emission of light into the surrounding countryside and to space.

The largest town modelled, Tralee in County Kerry, is the administrative centre of the county with over 5000 public lights within an area of 19 km$^2$. As a test case, we modelled the entire town, assuming that all public lighting was of the worst controlled type, i.e., the LPS 55 W prismatic lanterns present in the residential areas as described above. Our results are shown in Figure 2, where the towns from Paper One are shown as lines, while the new towns introduced in this paper are shown with a combination of lines and open symbols. It is readily apparent that the results for Tralee follow the same trend as the residential cases containing this type of lantern, but since all the lighting is set to this one type, the model output is at the upper bound of the previous results and can be taken as the worst case scenario in terms of light pollution. We note that the diffuse component for this town follows almost exactly a pure Lambertian distribution, although, as noted in Table 2, the fraction of the total upward emission escaping from individual areas varies from town to town and is always less than the obstruction-free case.

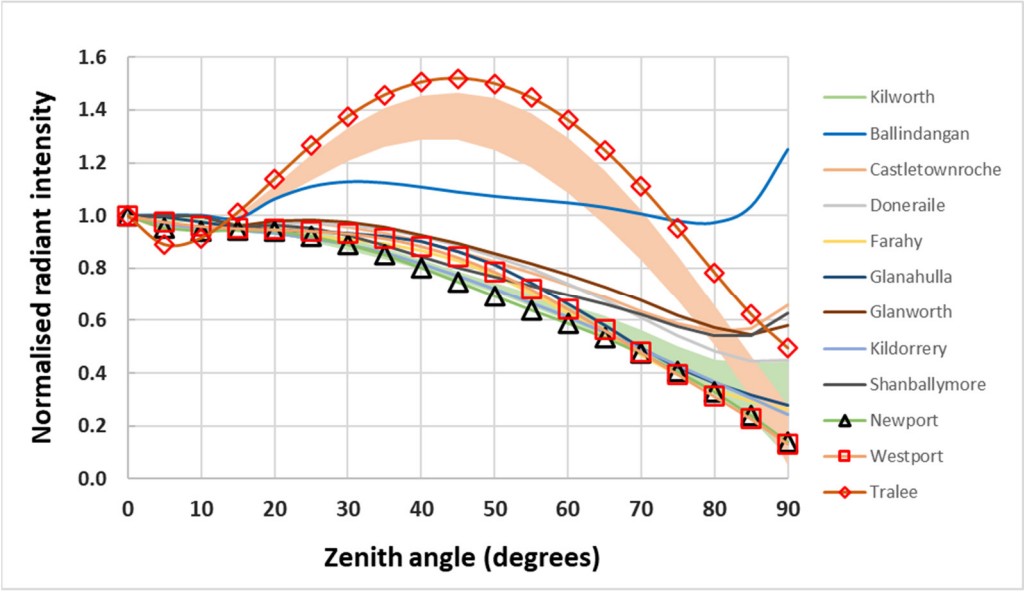

**Figure 2.** Azimuthally averaged total radiant intensity for the towns modelled in Paper One, together with model results for three additional towns shown by the lines and open symbols. The shaded regions show the range of values for the Dublin residential areas discussed above. See text for details.

Figure 2 also shows that the town of Ballindangan, which we reported in Paper One to have the worst-case light emission, also has the largest near-horizontal emission and remains the worst example of this in the combined towns sample, although it has lower emission to intermediate angles. This low angle excess is due to the presence of LPS 55 W prismatic lensed lighting combined with a very open environment consisting of relatively wide streets and detached single-storey dwellings, which permit the escape of near-horizontal light to the wider environment.

### 3.3. Dublin City

To compare our results with those for more urbanised areas, we analysed the central Dublin test area presented in Paper One. This area lies approximately one kilometre from the city centre and consists mainly of mid-rise commercial buildings set along wide streets with narrower streets off them. Within the area are a number of enclosed public parks, and there are also a few boulevard areas. The 1 m resolution LiDAR data presented in Paper One were obtained in March 2015 when the trees were in leaf, while the 2 m resolution data presented here were obtained between December 2011 and February 2012. The model output for these two datasets is indicated in Figure 3 by the curves marked "NYU area," as the original dataset was taken as part of a New York University programme. There is very

good agreement between the models based on the two seasonal datasets, with less than a 1% difference between the results in terms of both the total emission and its dominant diffuse component, and only a 3% increase in the weaker direct emission component between the summer and winter datasets. These results are, perhaps, to be expected as there is generally little tree cover aside from along the margins of the public parks where some lights are located, and also only relatively few cases where there are trees bordering the roadways.

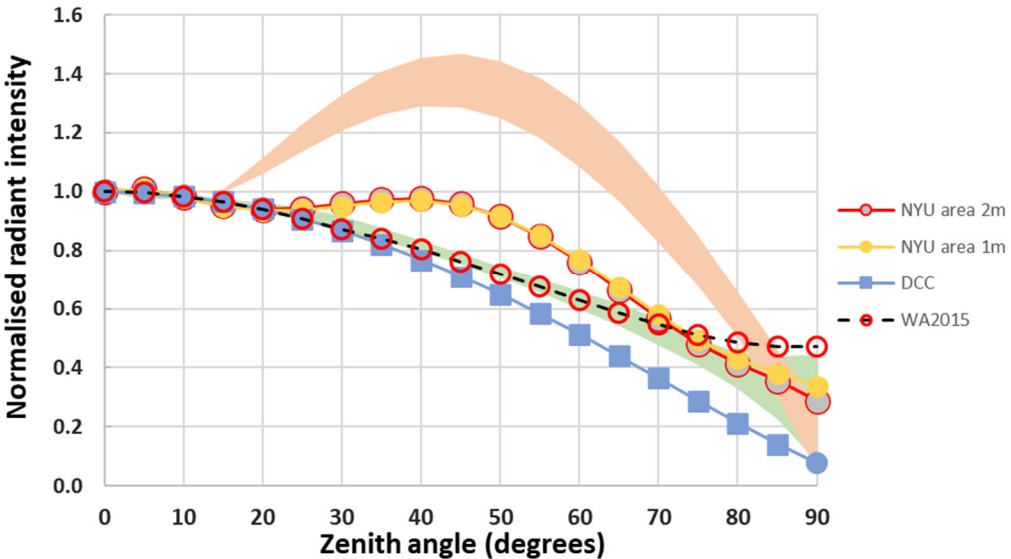

**Figure 3.** Total radiant intensity data for central city areas for summer and winter datasets (filled circles), both of which lie above the band of better-controlled residential lighting and are almost indistinguishable on this plot. Also shown is a model of the diffuse-only emission from the entire Dublin City Council region (squares), which is well approximated by a Lambertian model. The dashed curve with circles represents the model used for the 2015 World Atlas of light pollution. For comparison, the ranges of data for the residential areas described earlier are indicated by the shaded areas.

　　　In keeping with our previous observation that the biggest difference between models is due to the photometry of the predominant lantern type, we find that our result for the inner city area is roughly similar to that expected from one of the smaller residential areas with better-controlled lighting, consistent with the small proportion of 55 W LPS lanterns present in the inner city, which is dominated by higher wattage HPS lighting (see Table 2 for numbers). In the same Figure, we also plot the curve derived by Falchi et al. [8] from their inversion of global skyglow measurements, and it is heartening to see the close agreement between the two sets of results derived by different means, particularly in the case of the residential areas. We interpret this finding as follows: although other locations may have high-rise structures, it would be expected that the bulk of the emission at near-horizontal angles that contributes to skyglow in surrounding regions escapes from the low- and mid-rise areas with relatively few obstructions at low elevation angles. We also expect that the bulk of public lighting in the areas studied by Falchi et al. would now be of HPS type with better lighting control.

　　　Also included in Figure 3 is a test of our modelling approach to the entire Dublin City Council (DCC) area, amounting to over one hundred square kilometres and over 40,000 lanterns of both LPS and HPS types. The intention of this model was twofold: to see if the code would work efficiently with such a large area and the number of lanterns, and also to obtain an estimate of what the entire City Emission Function looks like for the case of well-controlled lanterns with no (or very little) direct emission above the horizontal. We thus restricted the model to a diffuse component only, and the model, including the generation of the base map of light pools around each light location and also the grid of shading models, ran in less than two hours on a modern 3.6 GHz CPU. It is notable that

this model is almost identical to a Lambertian distribution, although, as seen in Table 2, the emission to space is reduced relative to a model with no obstructions.

We calculated a version of the Li et al. [12] "blocking index" for the whole of the DCC area for the case of the two azimuthal angles from which the city of Dublin is viewed by the SUOMI satellite, viz. 99° and 289°. Our approach differs from that of the Li et al. paper in that we used the GIS plugin for shadow depth to generate a raster masked to show the visible areas from a given azimuth and elevation—see Appendix B for details. Having obtained masks for the required range of azimuths and elevations, a final output raster was generated for each azimuth in which each pixel contains the smallest elevation angle when that location first becomes visible, i.e., the blocking index value. Our approach has the advantage that it includes all obstructions over the entire mapped area simultaneously and is computationally efficient.

As the lit areas are also available, we can use these to weight the blocking index raster to select areas of relevance to the lit model. This has two advantages: firstly, that the numbers derived are for the individual locations, rather than random points along the street, and secondly, that better-lit locations receive a higher weighting, so the angle derived is more representative of the installed lighting. From our analysis, the minimum elevation angle required for a pixel to become visible is similar for both SUOMI azimuths and is equivalent to a zenith angle of 64° when averaged over the entire Dublin City Council (DCC) area. However, for an inner 2 km radius around the commercial heart of the city, the zenith angle is smaller by 6° (i.e., the elevation angle is larger) due to the presence of higher structures in this region. This finding indicates why the poorer lighting control of the 55 W LPS units is reduced towards higher zenith angles, as shown in Figure 3. This reduction is important as light emission in the range 80° < zenith angle < 90° is the dominant contributor to rural skyglow. In comparison, the relatively unobstructed emission at lower zenith angles contributes predominantly to the urban skyglow [13].

Although we do not incorporate an atmospheric scattering model in our calculations, we can give some approximate indications of potential light pollution through numerical integration of the light in the above angular ranges. One result is that we find that the ratio of light that can contribute in the "rural" to "urban" range is marginally lower in the LPS 55 W-dominated residential areas due to the increased proportion of light to lower zenith angles. When seasonal information is considered, this ratio increases from 4% in the winter data to 8% in the summer months, though it is also influenced by the restriction in light towards the zenith by tree canopies. For comparison, the ratio for towns (summer data only) ranges from 4% to 6%. Overall, we suggest that a "rural"/"urban" light pollution ratio of 6–7% during the summer months when foliage is present is a representative value for all locations examined.

### 3.4. Comparison with SUOMI Satellite Measurements

It has been pointed out by other authors ([12,14,15]) that the SUOMI satellite observations of cities taken at different nadir angles show discrepancies in the observed radiance, indicative of variations in light output with angles that differ from a simple Lambertian assumption. There are two general effects which can be observed: a light distribution in which the observed radiance initially decreases with increasing zenith angle, and a distribution where there is an increase with zenith angle. The former behaviour is ascribed to the effect of high-rise buildings obstructing emission towards the horizon, and the latter occurs where low-rise buildings are present. While there are relatively few high-rise buildings in Irish cities, we can compare our predicted low-rise area light to that observed for other such locations using the atmospherically corrected fits reported in Table 3 of [15].

For comparison with the Li et al. fits, in Figure 4 we plot our model results in terms of radiance, i.e., with a correction for areal projection effects, consistent with the way data from the SUOMI or NOAA-20 VIIRS DNB instruments are processed and reported. Similarly, we restrict the plotted range to a maximum zenith angle of 70° for consistency with the range of reported satellite data. Plotted like this, a true Lambertian response will

follow a horizontal line of unity as demonstrated, for example, by the case for the citywide diffuse-only model for Dublin (marked DCC in Figure 4). A reasonable upper limit is provided by the topmost line, which represents the results for the town of Tralee, modelled assuming the worst-case 55 W LPS lanterns. The plot shows that the majority of our results lie within the boundaries of the observed global low-rise areas, which suggests that our models have a generic application.

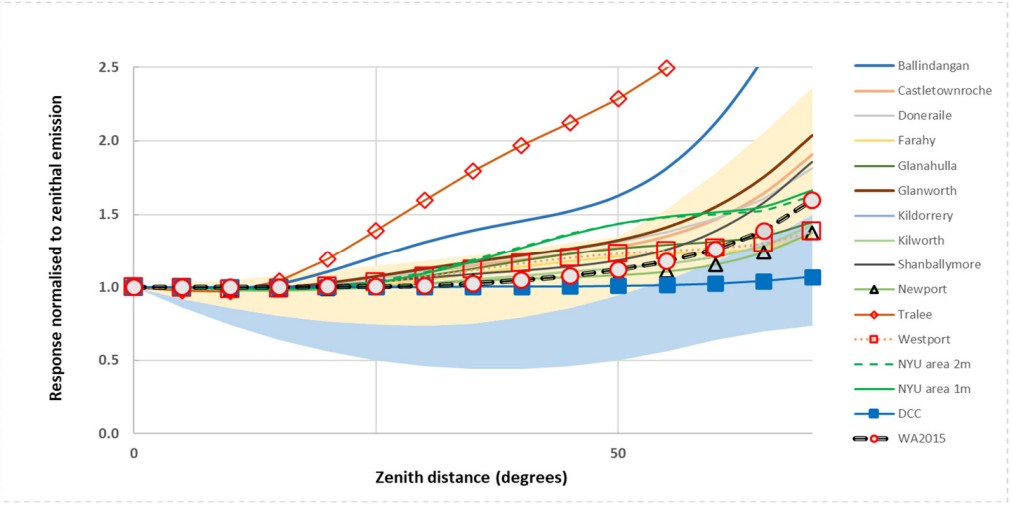

**Figure 4.** Model outputs for all urban areas reported in this paper are overlaid on the range of results for SUOMI VIIRS/DNB observations of worldwide cities reported by Li et al. [15]. The range of reported global values is indicated by the upper (cream-shaded region), which was identified as low-rise neighbourhoods, and the lower (light blue) shaded region was identified as high-rise neighbourhoods. For references to colour, see the online version of the paper.

*3.5. Seasonal Effects*

Although the impact of the high reflectivity of snow on light pollution has been discussed for other locations, snow events are relatively unimportant in the mild Irish maritime climate. However, the effect of snow on the ground is to increase the ground reflectivity, while leaving the direct emission unaffected, so we expect that for lanterns with relatively poor lighting control, the total detected output under such conditions will more closely approximate Lambertian emission when snow conditions are present.

Other than snow, a relatively large seasonal effect in temperate latitudes results from the change in vegetation levels over the year [16]. Such effects can also be examined in our models as datasets covering a range of dates are available. For the case of Dublin, the higher resolution dataset was obtained in the months of May 2013 and June 2018, while the lower resolution dataset was obtained in the period from mid-December 2011 to early February 2012. We return to the residential areas previously discussed as these consist of relatively open areas of two-storey housing with differing amounts of trees lining the roads and so might show seasonal variations. In Paper One, we noted that aggregating digital elevation data from 1 m to 2 m pixels does not affect the overall normalised emission results, and we appeal to this finding when comparing these two datasets to determine the effects of foliage cover on the emission function.

In Figure 5, we plot two extreme cases of behaviour due to seasonal effects, normalised to the winter zenithal emission in each case. The upper (winter) curves for both locations agree with a Lambertian model to a high degree, but there is a noticeable reduction of light output during the summer months, which differs between the sites. The region with the largest effect is a straight road with fully mature tree canopies up to 13 m high, while the lanterns are 10 m above the roadway—see Figure 6 for an illustration of the proximity of the tree canopies.

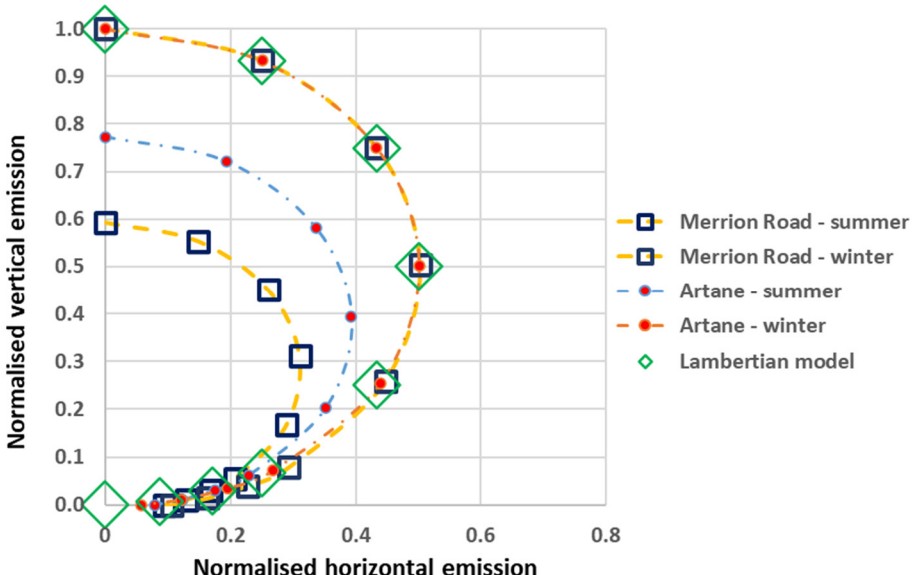

**Figure 5.** Emission functions of the diffuse (reflected) component alone for two extreme examples of foliage effects found in our residential sample areas. The curves have been normalised to the zenithal maximum of the winter emission for each location so that the relative decrease in summer months can be more clearly seen. The shape of the winter emission function is almost identical for both and has a Lambertian response, but the decrease due to summer foliage differs between the two locations. See text for details.

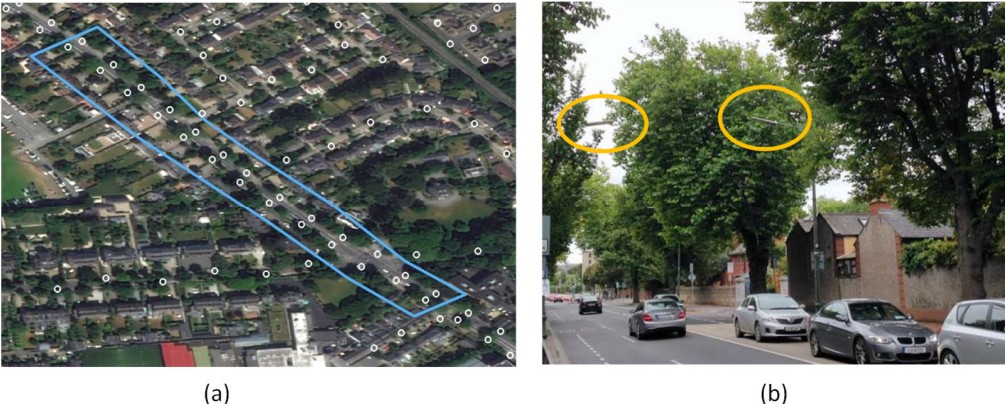

(a)                                          (b)

**Figure 6.** The Merrion Road residential location consists of a road lined by mature trees. In (**a**), the selected area is shown outlined on imagery from the QGIS ESRI base map, and this outline was also used for the extraction of data from both our raster output files and the calibrated ISS imagery. It can be seen from this plot that the locations of the light poles (open circles) quite often lie beside or in the canopy of deciduous trees. In (**b**), the location of two lanterns is outlined, indicating how close they are to the tree canopies, which rise several metres higher than the lighting poles.

A comparison of the seasonal results shows a maximum decrease in total output by 38% between winter and summer for the Merrion Road case, which compares favourably with the 31% value reported for tree-lined streets in Cambridge, Massachusetts [17]. Their result was obtained from sky visibility analysis derived from Google Street View imagery, though we note that our approach is much less computationally intensive. Representative results for other locations are listed in Table 2.

### 3.6. Azimuthal Photometry

Our work to this point has made use of azimuthally averaged direct emission values based on similarly averaged luminaire photometry. This averaged direct emission is com-

bined with the diffuse component to provide the total emission to space and the surrounding environment. The justification for this is that for relatively large numbers of well-distributed light sources, sampling of viewpoints to each light location should be (relatively) random, so azimuthally averaged values should be sufficient to provide relatively robust results. However, when we want to study small numbers of lights and/or a reduced set of orientations (e.g., when modelling a single straight road), then we need to consider that the lantern photometry is non-uniform in both azimuth and elevation and may, indeed, vary by location if lantern types differ. This is particularly true at low elevation angles (large zenith angles) as different light distributions are available for otherwise similar lanterns depending on the illumination required, e.g., on whether one or more carriageways of a road is to be illuminated. For most streetlights, the lighting distribution is chosen to produce an elliptical light distribution with its dominant direction along the street axis, i.e., in lighting photometric terms along the direction $0° < C < 180°$ (see Appendix A for an illustration).

We updated our model to include a photometric angular dependence by incorporating OpenStreetMap data to determine the azimuth of the road segment closest to each light. A suitable change of reference then provides the C-angle appropriate to a viewpoint from due North, which is easily modified to deal with any given line-of-sight azimuth (see Appendix B). Finally, by referencing the specific photometric tables appropriate to each lantern, we can determine the appropriate value of direct emission in our calculations. In Figure 7, we present a set of plots of the calculated total emission (diffuse + direct) for both azimuthally averaged and azimuth-specific values of direct emission to illustrate a range of conditions. By comparing the plots, it can be readily seen that at zenith angles below roughly 60°, there is relatively little difference between the two approaches, but for larger values, there is a variation in behaviour which is dependent on the lighting distribution and the relative orientation of the line-of-sight. As expected, the housing estate case with a more random distribution of orientations (Figure 6a) is closest to the averaged values, but relatively large deviations (and in opposite senses) are found for the other two cases, which model single lines of streetlights.

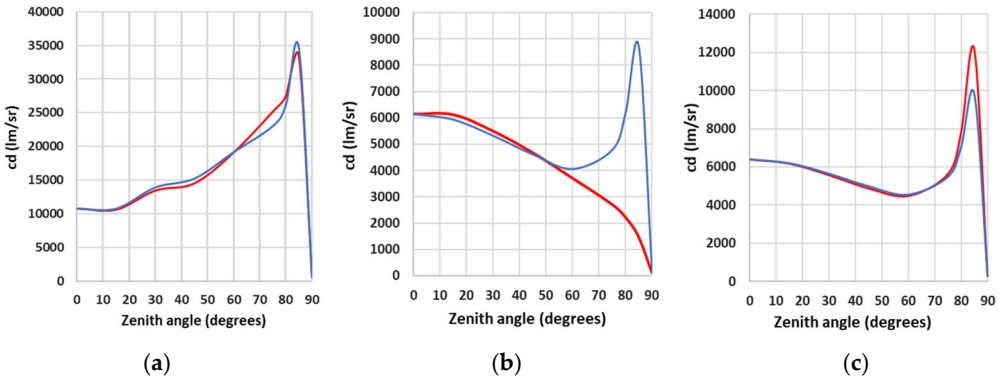

**Figure 7.** Three examples demonstrate the difference in total emission between the use of azimuth-averaged values (shown as the light blue line in all plots) and azimuth-specific values (shown with a dark red line) for the case of (**a**) a housing estate with many street orientations; (**b**) a single road oriented roughly parallel to the line-of-sight; and (**c**) a single road oriented roughly perpendicular to the line-of-sight. See text for details. For references to colour, see the online version of the paper.

The results suggest that large-scale modelling can be approached using averaged photometry, particularly for near-nadir satellite observations, but there can be differences in detail when individual areas are intercompared. This is especially true when modelling or comparing observations of areas of differing geometry and/or complexity, particularly in the case of light emission at near-horizontal angles appropriate to health and environmental impact studies.

### 3.7. Quantitative Comparison to ISS Data

As noted earlier, we located an image of Dublin (ISS045-E-170140) taken from the International Space Station in winter 2015, which had both good resolution (≈16 m/pixel) and was roughly contemporaneous with both the digital elevation and public lighting data. This image was georeferenced, calibrated to radiance units and corrected for atmospheric absorption and scattering by Dr. Alejanandro Sanchez de Miguel. We used the radiance-calibrated images in the G (green) band as these are roughly coincident with the photopic band used for determining our model radiances and hence facilitated comparison with our model output. In addition, by incorporating the known image corner coordinates into the NASA image footprint calculator spreadsheet, we could determine the location of the viewpoint relative to the ground and hence the appropriate azimuth and nadir angle to model (https://eol.jsc.nasa.gov/SearchPhotos/Low_Oblique_301_Locked.xls (accessed on 3 June 2023)).

As test areas, we used the Dublin residential locations discussed above, which we knew to be dominated by LPS emission. By using vector shapefiles, we were able to extract both model and observed radiances for each of the residential areas and compare them after converting both sets of measurements to watts per steradian. As before, we used a uniform reflectance of 10% for all areas and did not adjust the model output to account for any differences in environment or orientation beyond the selection of the appropriate azimuthal photometry for the observation geometry. The data can be seen in Figure 8, together with the best-fit linear model to the data. From the correlation coefficient of 0.93, it can be seen that there is a very good agreement between the model output and the observations. While our model results are, in general, slightly offset to higher values than observed, we note that we have not attempted to account for degradation in light output due to the total failure of individual lamps nor for the progressive decrease in output due to lamp degradation and/or dirty lantern lenses; in technical terms, the maintenance factor has been set to unity for all locations.

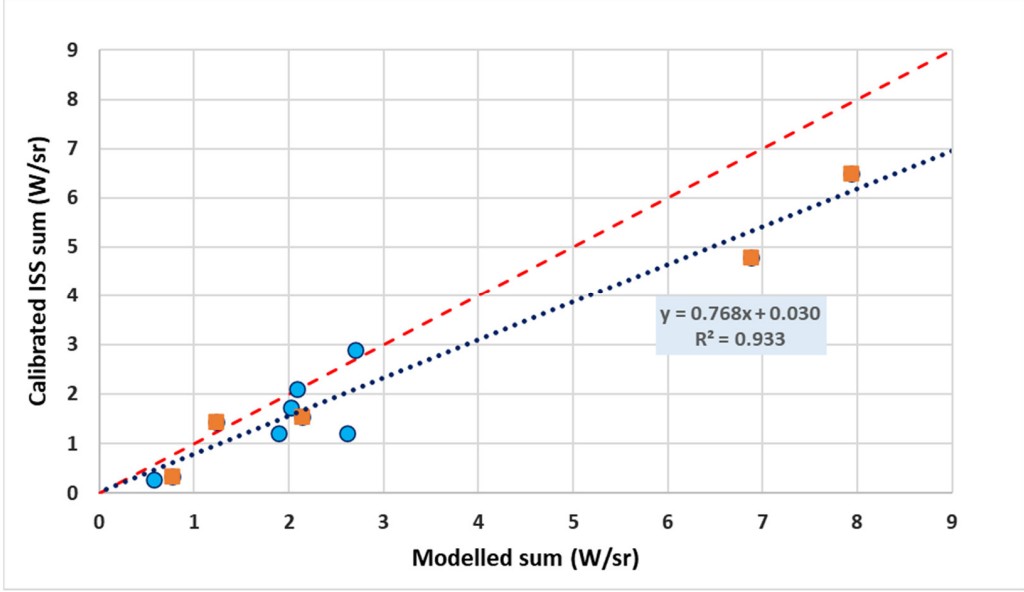

**Figure 8.** Comparison of the extracted output from radiance-calibrated ISS imagery with model predictions for the same areas. Orange squares represent residential areas with LPS 55 W lanterns, and blue circles represent other LPS areas. The dotted line shows the best-fit linear model to the whole dataset, while the dashed line illustrates a line of unity slope. See text for more details. For references to colour, see the online version of the paper.

This approach to testing our model makes use of the near-monochromatic output of LPS lamps present at the time of observation and modelling. Our approach would be

more complicated to apply with broad-spectrum lighting sources or for detectors with different spectral responses or, indeed, panchromatic imagers such as VIIRS DNB, even without consideration of the more complex atmospheric transfer for broadband spectra. However, we believe that this result is the first such detailed comparison between model and observation and that it vindicates our overall approach.

## 4. Discussion

We have presented a collection of results from our modelling of low- and mid-rise locations in Ireland using realistic photometric data. From our modelling, we find that the main difference in normalised emission function between low-rise locations is due to lantern photometry, with obstructions such as foliage causing a secondary effect. Our results are generally similar to the light distributions found for similarly low-rise developments worldwide, based on SUOMI satellite observations. As the range of Irish environments is limited due to relatively low building heights, we cannot comment on how our model would reproduce the results for the high-rise urban centres found in other countries, but we are currently working to address this using appropriate international data and hope to report on this shortly. While details will necessarily vary from location to location, the generality of the low-rise result is important for understanding the propagation of artificial light at night and also for making inferences from satellite data for similar locations worldwide. An example of a possible application would be informal developments in theGlobal South, though more work needs to be conducted to determine the fraction of light emitted upward for those locations. We also verified that the emission function developed from skyglow data by Falchi and co-workers is, in general, a good approximation to our results [8].

A related finding is that the production of a VIIRS Black Marble near-nadir output product based on data taken within 20° of the nadir is a good choice in terms of limiting the variation between observations and also between sites when correcting for nadir angle [18]. Although the resulting mean radiance does not directly enable the true value of emission to be determined, processed data will, at least, be consistent in that the assumption of a Lambertian dependence is closely met over the limited range sampled. With the continued move to better-controlled lighting, this should only improve, as has been shown in data taken with the VIIRS/DNB instrument, although observations with panchromatic detectors need to be treated with caution [18].

We have demonstrated and quantified the effect of seasonal foliage on the light emission from a range of areas and show that it is consistent with that found independently for a similar area using a different technique. Furthermore, the accuracy and wide-area coverage of digital elevation data enable such calculations to be performed quickly and completely and are much less computationally intense than the approach of Li and co-workers [12].

Development of our model to use more detailed photometric data to calculate the radiance towards a single point-of-view produces results in good quantitative agreement with calibrated space imagery, suggesting that our models can be developed to enable approximate calibrations for public lighting as imaged with ISS data, with appropriate consideration of atmospheric corrections.

Our results have implications for other ALAN workers, including those working in the fields of environment and health studies. The use of satellite imagery to determine the nature and amount of night-time light currently depends on assumptions of the light distribution at low angles that may vary markedly from simple assumptions. For work that involves the study of ALAN at a distance, such as quantitative satellite observations and long-range environmental impacts, radiative transfer and the treatment of different spectral sources will need to be included, as has been attempted by, e.g., [7].

As in Paper One, we have focussed on modelling public lighting, which accounts for all or a large part of the emission in the regions modelled. While this is not a complete picture, it enables us to compare the influence of environments directly without additional

complications. For the residential areas, as noted earlier, we used the ISS data as a guide to the selection of public lighting-dominated emissions. In addition, we are working on adding additional lighting sources to our model, such as floodlighting, in order to more completely represent other significant sources of light pollution and, in particular, of light that peaks close to the horizontal and hence will be of importance in terms of its impact both within urban areas and also on the peri-urban environment.

**Supplementary Materials:** The following supporting information can be downloaded at: https://www.mdpi.com/article/10.3390/rs15122973/s1, Files: S1_README.txt, S2_Generate_Hillshade_files.py, S3_Lighting_raster_models_and_analysis.r, S4_Generate_lantern_azimuth_instructions.pdf; Table S1: Lantern_photometry.csv.

**Author Contributions:** Conceptualisation, B.R.E.; Methodology, Software and Validation, B.R.E., X.Y. and K.P.; Formal Analysis, Investigation, Resources, Data Curation, B.R.E.; Writing—Original Draft Preparation, Review and Editing, Visualisation, B.R.E.; Supervision, Project Administration and Funding Acquisition, B.R.E. All authors have read and agreed to the published version of the manuscript.

**Funding:** We gratefully acknowledge funding provided by the Sustainable Energy Authority of Ireland (SEAI) under grant 22/RDD/868.

**Data Availability Statement:** All data other than the 1 m resolution data for Dublin City are publicly available from the Irish government web portal https://data.gov.ie (accessed on 3 June 2023). LiDAR data from this site is © Government of Ireland. These datasets were created for and are the property of the Office of Public Works (OPW), and this copyrighted material is licensed for re-use under the Creative Commons Attribution 4.0 International licence. Commercial elevation data for our study areas was obtained by Bluesky (https://www.bluesky-world.ie/ (accessed on 3 June 2023)) and shared through the kind assistance of Gerald Mills of University College Dublin. Open Street Map data used in this work is © OpenStreetMap contributors and available via. an Open Database License from https://www.openstreetmap.org (accessed on 3 June 2023). Esri imagery utilised in the production of this work was obtained via. QGIS QuickMapServices-ArcGIS® software by Esri. ArcGIS® and ArcMap™ are the intellectual property of Esri and are used herein under license. Copyright © Esri. All rights reserved. ISS imagery was provided by the Earth Science and Remote Sensing Unit, NASA Johnson Space Center (https://eol.jsc.nasa.gov/SearchPhotos/ (accessed on 3 June 2023)), and our thanks go to Alejandro Sanchez de Miguel of Complutense University of Madrid for providing the radiance calibration. We made use of the EU Corine database (https://land.copernicus.eu/pan-european/corine-land-cover/clc2018 (accessed on 3 June 2023)) which is © European Union, Copernicus Land Monitoring Service 2018, European Environment Agency (EEA). Figure A1 is adapted from Figure 2 of Skarżyński, Żagan and Krajewski (2021) available under Creative Commons License CC BY 4.0. Our code is undergoing improvements to automate it further as well as make it more user-friendly: contact the authors if you wish to obtain the current version.

**Acknowledgments:** We acknowledge Pat Caden, Executive Engineer (retired), Dublin City Council, for providing the lighting database for the city, and also the Irish Road Management Office (RMO) for providing data for the other areas outside Dublin City. We also thank Alejandro Sanchez de Miguel for calibration of the ISS imagery. Our thanks also go to Peter McEnroe for his undergraduate project work in which he identified suitable urban test areas and produced the initial shapefiles. Finally, we thank the three referees for their careful reading of the manuscript and their suggestions for improving the text.

**Conflicts of Interest:** The authors declare no conflict of interest.

## Appendix A

Lantern photometric measurements provided by manufacturers contain data in both the horizontal and vertical plane, and a realistic model of light output should take this into account, particularly where the numbers of lanterns are small so that statistical averaging of viewing angles to different street locations does not apply. In particular, linear features such as roadways result in a distant point-of-view sampling only a limited number of azimuth angles with respect to the lanterns.

Due to reduced obscuration at higher elevations (lower zenith angle), a wider range of viewpoints is possible, though again controlled by the axis of the roadway. In the C-Gamma system used for photometry and illustrated in Figure A1, the origin of the coordinate system lies along the road axis hence an elegant solution to the problem of obtaining lantern azimuths seen from any given viewpoint can be obtained by utilising OpenStreetMap (OSM) data. Using the OSM database, it is possible to download highway information for a given area and to obtain the alignment of a road segment in the vicinity of a given lantern. The azimuth from each lantern to the closest road segment for a large number of lanterns can then be determined in a matter of minutes. As the shortest distance line is perpendicular to the road axis, the lantern orientation (in photometric terms, the C-angle) can be easily found from this number as the origin is at right angles and to the right of this line (see Figure below). By tagging each lantern in the database with this azimuth, the orientation of a lantern to the line-of-sight is easily found by adding the given observation azimuth to the stored lantern value.

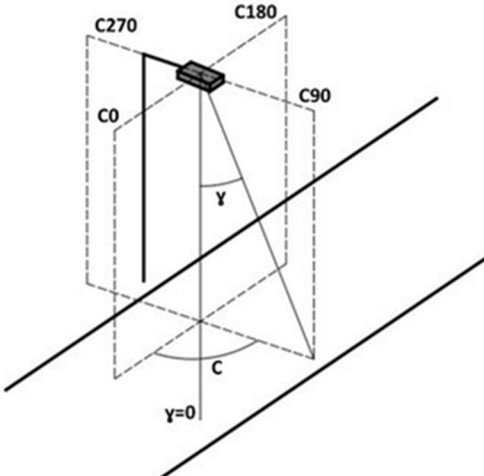

**Figure A1.** Diagram illustrating the definition of the (C, γ) angle definitions used in streetlight photometry. The Figure is adapted from that found in Figure 2a in [19]. by available under Creative Commons License CC BY 4.0 The original paper is available at: https://www.researchgate.net/publication/353590648_LED_Luminaires_Many_Chips-Many_Photometric_and_Lighting_Simulation_Issues_to_Solve (accessed on 3 June 2023).

**Appendix B**

Li et al. have made use of a visibility angle for each pixel location based on the blocking nature of buildings [12]. In their approach, they calculate the minimum elevation angle required to see the sky from the ground for all azimuths around every pixel location out to a user-defined radius. This is a computationally intense calculation with large rasters, and, in our case of the Dublin City Council, data amounts to nearly thirty million pixels. We have developed a more efficient GIS-based approach to calculate results for all pixels in the image based on casting shadows. For this, we make use of the QGIS Shadow dept plugin to generate shadow masks for a range of elevation angles (https://qgis.org accessed on 3 June 2023) This plugin generates a shadow raster for which the value at each pixel is the depth below a sharp shadow. We then process this output using a routine in the R GIS software (https://www.r-project.org/ accessed on 3 June 2023). By mapping the range of values to a binary mask—i.e., translating to a simple image illustrating locations lying above or below the shadow line—a virtual raster image can be efficiently generated for the entire image in one pass. The outputtakes into account all objects including tall buildings far away from the reference location or, for example, relatively low obstructions such as trees for sightlines near the horizontal. As the code runs, a check is made of the elevation angle at which a given point becomes visible, i.e., when the depth below the sightline ceases to

be negative, and that angle is recorded in each pixel of the raster frame that is written out on completion of the run.

Since the viewpoint of the SUOMI and NOAA-20 satellites is of interest in light pollution studies, and these satellites have a restricted range of azimuths from any given location, it is possible to efficiently generate a shadow map for a range of elevation angles for the entire image at one time. In our implementation, we calculated a grid of models for the relevant azimuthal angles (mean angles of 99° and 289° degrees for the Irish case) at 5° elevation increments. We can then access the location of individual lights to determine the viewpoint statistics or even use an OpenStreetMap vector road map to calculate the minimum visibility angle along all roads within the area of interest.

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
