# Peer review of "Real-World Urban Light Emission Functions and Quantitative Comparison with Spacecraft Measurements"

_remotesensing, doi:10.3390/rs15122973_

Round 1
Reviewer 1 Report
Review 'Real-world urban light emission functions and quantitative comparison with spacecraft measurements' by Espey et al. (2023)
Summary
The authors present a novel way of determining the (city) emission function, one of the most valuable inputs for skyglow and light pollution modelling. They do so by considering the total light distrubution of individual lighting sources, define blockings and diffuse portions assumpted from Irish in-situ conditions and calculate results compared to satellite imagery. Data are comparable, which show that Lambertian emissions can be the matter of choice at some theoretical build-ups, well defined by the authors.
Judgment
The authors present new theories of high quality and proved scientifically. In current existing light pollution models, it is of high importance to come closer to small scaled build-ups while keeping computational time relatively low. The emission function is of high importance for this and the way authors introduce its determination could be of high relevance for future works. The paper is therefore highly suitable for publishing in Remote Sensing. However, I would be happy if authors could address some questions which are unclear to me and could also be important for other readers. Maybe some parts in the paper could be better described in more detail. Very critical is the use of citation numbers, which are wrong starting with number 7 and higher. I therefore recommend a minor revision. My questions and comments are listed below.
Comments
-> general: There seems to be a critical problem with citations. In the reference list, citation 8 is not cited while citation 8 in the text seems to refer to citation 9, 9 -> 10, 10 -> 11, etc. Please review all citations to be clear of their correctness.
-> p. 1, ln 32: same problem with citation 'Falchi et al.'
-> p.2, ln. 76/77: Please include a reference for choosing 10% of surface reflectivity. In Kotthaus et al. (https://www.sciencedirect.com/science/article/pii/S0924271614001233) reference values for urban areas are closer to ~20-30%.
-> p. 2, ln. 81: Please include a citation/link for LiDAR DEM datasets.
-> p. 3, Table 2: I miss, how the exact values were determined? Are those assumptions or coming from real data? Maybe one or two sentences describing the numbers could be added in the text.
-> p. 4, ln. 137: Please include a citation/link for ISS imagery origin.
-> p. 5, FIgure 1: The legend is hardly readable, maybe it can be expanded or enlargened.
-> p.7, Figure 3: The grey line (NYU 2m) is hardly distinguishable from the yellow one (NYU 1m). Maybe it would be better to use a black dotted line for NYU 2m to highlight is similarity compared to the NYU 1m. Accordingly, WA2015 should be changed.
-> p. 8, ln. 304: typo '...where the where there...'
-> p. 9, Figure 5: I know what the authors want to show with the plots, however to unexperienced readers it could be irritating that x- and y-axis both show 'fractional emission'. Maybe the authors could include some details in x- and y-axis description or use a simple timeline plot?
-> p. 10, ln. 375: typo '...Figure Figure...'
-> p. 12, Figure 7: Why does the data analysis here only include 10 data points? Was it not possible to use more?
-> p. 12, ln. 472: typo in citation [187]
-> Data Availability: Is the model code used by the authors available? I would be happy if authors could include some statement about this in the Data Availability statement.
Reviewer 2 Report
This paper is an excellent extension of the work reported in Espey 2021 on the detailed modeling of the angular distribution of emission of artificial light at night from urban areas. It is written understandably, the assumptions are clearly stated, and the modeling based on those assumptions compared and verified. The modeling approach has particular value because it can build on public datasets for built topography and makes just the right simplifying assumptions for computational efficiency. I have no critical substantive comments on the contents as presented, and recommend the paper for acceptance. The authors may wish to consider the following forward-look comments (and one minor issue with references) before submitting their final version.
Because the details of the urban emission function are so important to local and regional assessments, this tool is likely to become in demand, as it is expanded further to more challenging situations. One would be higher contrast shielding from taller buildings in more concentrated urban centers. The basic approach expounded here and in Espey 2021 seems likely to be able to accommodate that challenge, and it will be interesting to see if the conclusion from this paper continues to hold that the emission function is dominated by the photometric profiles of the fixtures rather than the local shielding.
The other is moving beyond the relatively simple calibration process of the nearly monochromatic sodium lamps to that for the broad-spectrum LEDs in more and more common use. On the one hand, the newer fixtures tend to have a much lower fraction of direct radiation above horizontal. An obvious issue is the correction for spectral sensitivity of the remote sensor, such as the strong cutoff of the VIIRS instrument that records no blue light. The more subtle point is that it may not be possible to neglect atmospheric scattering properties, particularly when modeling impacts on somewhat remote sites of special protection, such as nature preserves or astronomical observatories, where scattering from somewhat distant sources will dominate the contribution of ALAN to zenith sky brightness. With a heavier blue light contribution, the amount of scattering will then depend on the aerosol content. The statement in the discussion section starting on current manuscript line 478 is therefore true that good quantitative agreement was obtained, but that is primarily true for (nearly) monochromatic sources and light pollution in the vicinity of those sources. To the extent that there will be interest in regional protection against artificial LED illumination, including special remote areas, I’m not sure that the need for atmospheric corrections will be obviated.
At the copy-editing level, the references seem to be off by at least 1 ordinal number, starting at low numbers – was a reference early in the text inserted and not propagated?
Reviewer 3 Report
TITLE: Real-world urban light emission functions and quantitative comparison with spacecraft measurements
This paper is mainly devoted to the problem of light pollution modeling. Although it concerns the essential and contemporary issue linked to the light emission functions from residential areas, it can not simply be published in its present form. There are many inaccuracies in this study. The text is chaotic, and it is very tough to understand the content and the general outcome of this research. Therefore, it is highly recommended to organize the presentation of the conducted research and make the necessary corrections:
- The abstract must contain the main result from this study.
- The introduction must include the research background (proper literature study).
- The aim of the study is presented in two places (L.25, L.52-55). It must be unified.
- The simulation models, assumptions, and software must be clearly defined and described. It is unclear how the simulations, calculations were performed and the exact data used to obtain the results.
- Such long descriptions of figures must be avoided. Figures must be explained in the text and referenced. All axis must be described and contain units. Moreover, the figures' quality (sharpness) must be improved (fig. 6(1) and fig. 6 6(3). There are two figures, no 6. So, the additional check of the numbering correctness is required. For Fig. 5, it is good to provide the line of Lambertian distribution to show the reader that the differences are negligible.
- Duplicating the same information should be avoided, e.g., L.76-77 and L.85-86. Moreover, this information requires a proper reference.
- It is unclear what "Paper 1" is, but it appears in several places in the text. It must be clarified.
- The description of the figure in the appendix should contain the reference.
- All symbols and abbreviations must be explained, for instance, by adding an adequate list at the text's beginning or end.
- L.391-394 Light distribution of road luminaires is specific. It should be as wide as possible in the C0-C180 to provide a significant distance between the poles and much narrower in C90-C270 according to the road width, pole height, and lighting class. Please clarify it.
- L.334 What "relatively poor lighting control" means. On what basis was it assumed that snow conditions make the emission much closer to the Lambertian?
- There is no conclusion section. It needs to be clarified what is the outcome of this research.
I am not a native speaker, but I know English well. However, the language used in this article was difficult for me to understand because of very long sentences.
Round 2
Reviewer 3 Report
Dear Authors,
Thank you very much for the improved version of your paper. There are still issues with which I can't entirely agree. But I accept your point of view. The explanations I received are enough, and the changes improved the text.
Please include the main research conclusion in the abstract.
All the best!
The text is still hard to read. Language correction is required.
Author Response
We appreciate the comments of the referee and have amended the abstract to provide an overall conclusion.
